# From Spores to Solutions: A Comprehensive Narrative Review on Mucormycosis

**DOI:** 10.3390/diagnostics14030314

**Published:** 2024-01-31

**Authors:** Sara Palma Gullì, Vinaykumar Hallur, Pratibha Kale, Godfred Antony Menezes, Alessandro Russo, Nidhi Singla

**Affiliations:** 1Department of Medical and Surgical Sciences, Magna Graecia University of Catanzaro, 88100 Catanzaro, Italy; sarapalma08@gmail.com (S.P.G.); a.russo@unicz.it (A.R.); 2Department of Microbiology, All India Institute of Medical Sciences, Bhubaneswar 751019, India; 3Department of Clinical Microbiology, Institute of Liver and Biliary Sciences, New Delhi 110070, India; drpratibhapgi@gmail.com; 4Department of Medical Microbiology & Immunology, RAK College of Medical Sciences, Ras Al Khaimah P.O. Box 11172, United Arab Emirates; godfred@rakmhsu.ac.ae; 5Department of Microbiology, Government Medical College and Hospital, Chandigarh 160030, India; nidhisingla76@gmail.com

**Keywords:** mucormycosis, zygomycosis, diagnosis, management, breakthrough mucormycosis

## Abstract

Mucormycosis is an infrequent but fatal illness that mainly affects patients with uncontrolled diabetes mellitus, diabetic ketoacidosis, solid and hematologic neoplasms, organ transplantation, chronic steroid intake, prolonged neutropenia, iron overload states, neonatal prematurity, severe malnutrition, and HIV. Many cases were reported across the world recently following the COVID-19 pandemic. Recent research has led to a better understanding of the pathogenesis of the disease, and global guidelines are now available for managing this serious infection. Herein, we comprehensively review the etiological agents, pathogenesis, clinical presentations, diagnosis, and management of mucormycosis.

## 1. Introduction

Mucormycosis is an uncommon but stern fungal illness that has gained increasing attention recently because of the large number of cases reported during the COVID-19 pandemic. This review provides a comprehensive overview of etiological agents, clinical presentations, pathogenesis, laboratory diagnosis, management, breakthrough infections, and pediatric mucormycosis. Understanding the multifaceted nature of this disease is critical for effective diagnosis and management, making this review a valuable resource.

## 2. Etiological Agents Causing Mucormycosis and Their Taxonomy

The etiological agents causing mucormycosis belong to the kingdom: Fungi, phylum: Mucormycota, subphylum: Mucormycotina, and order: Mucorales. The term zygomycosis, encompassing all fungi that produced coenocytic hyphae and zygospores, has been abolished in modern taxonomy. Further, *Conidiobolus* and *Basidiobolus* are no longer grouped with the mucoralean fungi as the clinical manifestations of entomophthoromycosis and mucormycosis are different, and the grouping was not tenable in molecular phylogeny [1]. Human pathogenic Mucorales belong to 12 genera viz. *Actinomucor*, *Apophysomyces*, *Cokeromyces*, *Cunninghamella*, *Lichtheimia*, *Mucor*, *Mycotypha*, *Rhizomucor*, *Rhizopus*, *Saksenaea*, *Syncephalastrum*, and *Thamnostylum*. Thirty-nine species within the above genera were found to cause human infections in the past, and recently *Cunninghamella arunalokei* was described as a novel agent of mucormycosis [1,2,3]. Almost half (48%) of all the cases were caused by *Rhizopus*, followed by *Mucor* (14%) and *Lichtheimia* (13%) in pre-COVID-19 times [2,4]. In a metanalysis on COVID-19-associated mucormycosis (CAM) by Ozbek et al. involving 958 patients that analyzed 233 cases with data on pathogens isolated, *Rhizopus* (70.4%) was still the commonest cause, followed by *Mucor* (21.5%) and *Lichtheimia* (4.7%) [5]. Geographical variations exist in the rank order of causative agents of mucormycosis; for example, *Apophysomyces* is the second most common cause of mucormycosis in India compared with *Lichtheimia* elsewhere [6]. Variations in environmental conditions are thought to lead to such variations. Etiological agents are also known to influence clinical presentation.

## 3. Pathogenesis

Mucormycosis, primarily an opportunistic infection, manifests in patients with underlying conditions such as uncontrolled diabetes mellitus, diabetic ketoacidosis, solid and hematologic neoplasms, organ transplantation, chronic steroid intake, prolonged neutropenia, iron overload states, neonatal prematurity, severe malnutrition, and HIV [7]. The precise sequence of events precipitating disease following the inhalation or implantation of the fungus into host tissue remains elusive, prompting the formulation of hypotheses grounded in experimental observations, mainly in rhino-orbito-cerebral and pulmonary mucormycosis. Ubiquitous in the environment, mucoralean fungi thrive in soil, decomposing organic matter, and animal dung [6]. Spores disperse into the air upon sporangial wall disintegration, reaching the upper respiratory tract and lungs. In immunocompetent individuals, spores are efficiently transported down the pharynx through the ciliary escalator mechanism and excreted via stools.

Local factors, including retentive anatomical niches, mucous stasis, and dehydration, impede adequate pathogen clearance. Tissue damage in diabetes or after cytotoxic chemotherapy exposes laminin and collagen IV, binding mucoralean spores and rendering expulsion difficult, leading to internalization and germination [8]. Germinated spores invade host cells, secreting lytic enzymes and toxins such as mucoricin, inducing tissue thrombosis and necrosis [9,10]. In healthy individuals, innate and adaptive immunity and platelets prevent infection [11]. Epidermal growth factor pathways additionally prevent cellular extrusion, safeguarding against mucormycosis and maintaining skin and mucocutaneous tissue integrity [8]. Disease development hinges on the interplay between host immunity and environmental exposure [12].

In type 2 diabetes (T2DM), elevated glucose levels induce cellular hypoxia and stress, releasing free radicals, fatty acids, and inflammatory cytokines. This creates a proinflammatory state associated with phagocytic cell dysfunction, glycation of proteins sequestering iron with better iron availability to fungi, and tissue infiltration by M1 macrophages incapable of eliminating mucoralean spores [13]. Diabetic ketoacidosis (DKA) further lowers blood pH, and *R. arrhizus* produces ketoreductase, utilizing ketones to enhance its development. These factors affect both the host milieu and the pathogen with surface translocation of GRP78 to host nasal epithelial cells and increased expression of spore coat proteins (cotH3 and cotH7) in the fungus. Mucoralean spores bound to exposed laminin and collagen IV in the nasal or sinus niche activate calcineurin pathways, germinate, and germlings bind to GRP78, invade the nasal epithelium, and express mucoricin-inducing apoptosis and necrosis. The toxin increases vascular permeability, facilitating dissemination via blood. Neutrophils recruited to the infection site kill hyphae but release mucoricin, perpetuating tissue necrosis and inflammation and impeding further phagocyte influx and antifungal delivery [10].

While GRP78 expression increases on nasal epithelial cells in uncontrolled diabetes and DKA, integrin β1 expression rises on alveolar epithelial cells in patients with hematologic malignancies [13,14]. The spore coat protein CotH7 binds integrin β1, activating the epidermal growth receptor, leading to cellular invasion, tissue destruction, and pulmonary mucormycosis [14]. There is no evidence that COVID-19 directly leads to COVID-19-associated mucormycosis (CAM). Instead, a combination of factors, including exacerbated T2DM or DKA, immunosuppressive treatments like steroids aggravating T2DM, and an altered microbiome due to antibiotics like azithromycin, likely contributed to the alarming CAM epidemic [15,16]. This perspective aligns with most CAM cases involving the nose and paranasal sinuses [5,15,17].

## 4. Clinical Manifestations

Here, we discuss the clinical manifestations of mucormycosis depending on the anatomical site involved. Pediatric mucormycosis and breakthrough infections are discussed separately, as these are usually neglected and often occur in niche populations.

### 4.1. Rhino-Orbito-Cerebral Mucormycosis

Rhino-orbito-cerebral mucormycosis (ROCM) is the most common presentation of mucormycosis and occurred in 34% of patients in a global meta-analysis by Jeong et al. [4]. Uncontrolled diabetes or diabetic ketoacidosis is the most common risk factor for ROCM [18,19]. Infection is acquired following inhalation of spores, and the patient usually presents acutely with one or more combinations of the following symptoms: acute onset unilateral facial swelling with/without numbness or tingling, bulging of the eyes with or without loss of vision, headache, nasal stuffiness, and presence of eschar or ulcer in the palate or the nasal cavity. A black necrotic eschar in the palate or nasal mucosa, although present only in 50% of cases, is considered a sentinel sign for ROCM [20].

### 4.2. Pulmonary Mucormycosis

Pulmonary mucormycosis is the second common presentation of mucormycosis and occurs following inhalation of spores. Solid organ transplant, prolonged neutropenia, hematological malignancies, bone marrow transplant, and steroid use predispose a patient to pulmonary mucormycosis [4,19,21]. Patients with solid organ transplantation have the highest odds of developing pulmonary mucormycosis than other clinical syndromes [4]. Clinical presentation of pulmonary mucormycosis is nonspecific, with complaints like fever and cough, and more than a quarter (28.3%) of them can have hemoptysis mimicking bacterial pneumonia with rapid progression. Invasive aspergillosis is a close differential diagnosis, and radiological findings like reverse halo sign, pleural effusion, and presence of more than ten nodules are said to be more predictive of pulmonary mucormycosis than invasive aspergillosis [22]. *Cunninghamella* species are primarily associated with pulmonary manifestations among the causative agents.

### 4.3. Primary Cutaneous Mucormycosis

Primary cutaneous mucormycosis occurs following direct percutaneous inoculation of fungal spores and is not due to dissemination from another site. Consequently, patients with significant trauma like road traffic accidents and burns or minor trauma like injections have the highest odds of developing the disease [4,19,21]. Nonetheless, a meta-analysis found that infection occurred in 39.7% of patients without underlying risk factors [22]. As per this study, cutaneous mucormycosis frequently presents acutely and may rapidly progress, leading to gangrene or hematogenous dissemination. Classically, the infection presents as black eschar, but alternative words like necrosis, necrotizing infection, and blackish discoloration of underlying skin or ulcer have also been frequently used in the literature [23]. *Apophysomyces*, *Lichtheimia*, and *Saksenaea* are more frequently isolated from patients with cutaneous mucormycosis than members of other Mucorales [4]. Dissemination to deeper organs occurred in approximately 20% of patients in a review [21].

### 4.4. Gastrointestinal Mucormycosis

This disease has been chiefly described in Asia and occurs after ingesting food contaminated with mucoralean spores or exposure to infected devices used in hospitals [19,24]. It occurs in both immunocompetent and immunocompromised hosts and neonates [25]. Solid organ transplantation (SOT) in 52% of patients, followed by hematological malignancy in the remaining patients, were the underlying diseases among immunocompromised patients in an older review of 31 cases of GI mucormycosis [26]. In this review, intestinal mucormycosis was significantly associated with hematological malignancy compared with gastric involvement in patients with SOT. A study of 176 immunocompetent patients found that the disease involved adults and children, and involvement of the intestines (64.2%) was more common than the stomach (33%) [25]. The signs and symptoms of GI mucormycosis are nonspecific and consist of abdominal pain, upper or lower GI bleeding, fever, etc. [27].

### 4.5. Disseminated Mucormycosis

When the disease involves more than one noncontiguous site, it is called disseminated mucormycosis [4,19,21]. It occurs mainly in patients with profound immunosuppression and has the highest mortality compared with other clinical types. In a review of 851 cases of mucormycosis by Jeong et al., patients with major trauma (odds ratio 8.55), followed by solid organ transplantation (odds ratio 4.2) and hematological malignancies (odds ratio 3.8) had the highest odds of development of disseminated mucormycosis. It was reported in 13% of cases in the above study, and *Cunninghamella* species were significantly associated with disseminated mucormycosis [4,19,21].

### 4.6. COVID-19-Associated Mucormycosis

The highest number of CAM cases were recorded in India [5,15,17]. In a study on 958 CAM cases, DKA and steroid overuse were the commonest underlying risk factors [5,28]. All the clinical presentations mentioned above have been observed in patients with CAM. ROCM was the most common presentation worldwide in patients with CAM and had lower mortality (14% or more) compared with other presentations like pulmonary (70–80%), primary cutaneous (50%), gastrointestinal (76%), and disseminated mucormycosis (76%) [15,17,28].

### 4.7. Pediatric Mucormycosis

Pediatric Mucormycosis is rare but is increasingly reported, particularly in neonates where gastrointestinal mucormycosis manifests as necrotizing enterocolitis (NEC). It should be suspected in premature low-birth-weight babies with prolonged neutropenia and on treatment with multiple antibiotics [29]. Broad-spectrum antibiotics, steroids, and medical interventions disrupt gut flora predisposing to infection. Community-acquired infections may result from consumption of contaminated food. Immaturity of the immune system is an additional risk factor [27,29,30]. Mortality rate is very high, and many cases are identified postmortem [31]. Pediatric gastrointestinal mucormycosis is differentiated from classical NEC by the absence of pneumatosis intestinalis, poor response antibiotics, and widespread gut vessel thrombosis [30,32]. Roilides, in an extensive review, reported an analysis of 187 cases of mucormycosis in neonatal and pediatric age groups until 2014 and concluded that gastrointestinal presentation, extensive involvement, dissemination, and prematurity were hallmarks of neonatal mucormycosis and differed significantly from children of older age group [32].

Besides gastrointestinal mucormycosis, cutaneous disease is the second most common presentation among neonates [33]. In a study analyzing 157 pediatric mucormycosis cases by Zaoutis et al., cutaneous mucormycosis was the predominant manifestation involving males (64%) with a median age of 5 years [34]. In older children, clinical manifestations mirror those in adults characterized by rhino-cerebral and pulmonary involvement. Childhood malignancy, neutropenia, abdominal surgeries, or immunomodulation for autoimmune diseases are predominant risk factors, with diabetes mellitus being a minor risk factor in contrast to adults [32].

### 4.8. Breakthrough Mucormycosis

Breakthrough mucormycosis describes a clinical condition in patients who develop mucormycosis despite being on antifungal prophylaxis with drugs with or without activity against Mucorales. Voriconazole, a frontline antifungal for invasive aspergillosis, has no action against Mucorales, often leading to breakthrough mucormycosis in high-risk patients undergoing prophylaxis. This phenomenon, initially documented by Marty et al. in individuals with hematological malignancies on voriconazole prophylaxis, stems from a hypervirulent switch induced by epigenetic modifications in the fungus [35].

Breakthrough mucormycosis is not exclusive to voriconazole and has been reported with other antifungals, too, such as posaconazole, isavuconazole, echinocandins, and amphotericin B. Studies indicate a substantial 34% incidence of breakthrough infections with posaconazole in severely immunosuppressed patients with active hematological malignancies [36,37]. Kang et al. and Lebeaux suggest varying sensitivity to posaconazole among Mucorales and low plasma concentrations of posaconazole due to diarrhea and mucositis in these patients further contribute to breakthrough mucormycosis [37,38].

## 5. Laboratory Diagnosis of Mucormycosis

Traditionally, laboratory diagnosis of mucormycosis is based on histopathology as the gold standard, supplemented by microscopy and culture [39], as detailed in Figure 1. Emerging diagnostic methods aim to expedite mucormycosis detection.

**Sample collection:** Biopsies or surgically debrided tissue are the best samples for laboratory diagnosis. If a biopsy is impractical, nasal discharge, aspirates, scrapings, and lavages are alternatives. Samples for microbiological analysis should be sent in a sterile container in normal saline, with swabs discouraged due to potential confusion of hyphae with cotton fibers and superficial sampling, which may lead to a missed diagnosis [40]. The samples should be sent to the laboratory immediately at room temperature as Mucorales do not survive at low temperatures. In the microbiology laboratory, grinding of samples is not recommended as the hyphae are easily damaged. Instead, the samples should be cut into small pieces, teased gently, and subjected to microscopy and/or culture. Lavage samples must be concentrated by centrifugation before further processing [40,41].

**Histopathological examination:** Histopathology, the gold standard for diagnosis, detects typical broad ribbon-like, pauci-septate or aseptate with right-angle branching mucoralean hyphae that invade into tissue and blood vessels leading to thrombosis and infarction, a characteristic feature of mucormycosis [39,41]. Hematoxylin and eosin (H&E) staining shows tissue necrosis with neutrophilic infiltration and focal granulomatous inflammation [41,42,43,44]. Special stains like periodic acid–Schiff (PAS) and Grocott’s methenamine–silver (GMS) can aid in the visualization and differentiation of the hyphae [42,43]. Histopathology may, however, miss the sparse hyphae, and mucoralean hyphae may be mistaken as other fungal hyphae as these may appear crinkled in the tissue sections [39,43,44]. Recently introduced commercial assays based on immunohistochemistry using monoclonal antibodies against *R. arrhizus* help distinguish between aspergillosis and mucormycosis with 100% sensitivity and specificity. These assays are particularly useful when cultures are sterile. Immunohistochemistry assays are moderately recommended (B IIu) by ECMM/MSG ERC guidelines [7].

### Microbiological Diagnosis

**KOH mount:** KOH mount is a presumptive diagnostic method that involves treating tissue with 10–20% potassium hydroxide (KOH) and microscopic examination. Mucoralean hyphae exhibit coenocytic, right-angle branching, distinguishing them from the acute-angled, dichotomous branching of *Aspergillus* hyphae, which also causes invasive sinusitis with angioinvasion. Fluorescent optical brighteners like Calcofluor or Blankoflour enhance hyphal visualization, binding specifically to the fungal cell wall component, β1-3, β1-4 glycoside chain of chitin [39,45]. Combining these stains with KOH increases sensitivity, rendering hyphae bluish green under UV light [39,46]. Direct microscopy, though inexpensive, is an essential tool for rapid fungal hyphae identification, providing a presumptive diagnosis. The ECMM/MSG ERC guidelines recommend direct microscopic examination for mucormycosis diagnosis alongside histopathological examination [7]. However, the failure to identify individual fungi and subjective hyphae identification is a drawback.

**Culture:** Fungal culture is an essential diagnostic modality for Mucorales identification to genus and species level and antifungal sensitivity testing [39,41]. Conventional culture methods, utilizing Sabouraud’s dextrose agar (SDA) and potato dextrose agar (PDA), involve inoculating and incubating samples at 25–37 °C. Usually, Mucorales form cottony fluffy colonies with black spores, creating the characteristic “salt and pepper appearance” within 48–72 h. Identification relies on lactophenol cotton blue mount (LPCB), assessing the presence or absence of rhizoids, branching pattern of sporangiophore, sporangium size/shape, and additional structures like columella, apophysis, and collarette. Zygospores, if present, also help in species differentiation [39,41,47]. Certain species, like *Cokeromyces recurvatus*, exhibit thermal dimorphism with a yeast phase at 37 °C and mold phase at 25 °C and should be considered while handling samples from patients with suspected mucormycosis [48]. Matrix-assisted laser desorption ionization time-of-flight mass spectrometry (MALDI-TOF) aids species identification yet may miss rare isolates due to database limitations [49]. Ruling out contamination is crucial, and correlation with clinical and radiological features is essential. However, culture yields are often low (50% sensitivity) due to various factors such as sample collection, storage at 4 °C, and tissue grinding [7,41].

**Newer diagnostic assays:** Antigen detection assays like serum galactomannan or beta-D-glucan are well established for invasive fungal infections due to *Aspergillus* or *Candida* and help in their rapid diagnosis in patients with risk factors [50,51]. Unfortunately, no commercial antigen detection assay is available for mucormycosis, and both the above assays fail to detect Mucorales, see Figure 1. Numerous attempts have been made to standardize specific molecules, such as evaluating the monoclonal antibody 2DA6, which reacts with the fucomannan antigen of *Mucor* spp., as conducted by Burnham–Maurish et al. [52]. Initially, they assessed a sandwich ELISA and developed a lateral flow immunoassay (LFIA) for Mucorales cell detection in clinical samples, evaluating the assays in mice models [52].

**Molecular methods:** There is an urgent need for an accurate molecular test to identify Mucorales directly from the samples, given the unreliability of cultures and the absence of serological tests. Molecular methods represent promising new tools for this purpose, involving the identification of fungal DNA directly from blood, respiratory samples, tissues, or urine. In patients with risk factors, these tests demonstrate enhanced utility for early diagnosis and treatment, as evidenced in a study by Legrand et al., where the detection of circulating Mucorales DNA (cmDNA) in severe burns patients helped in early diagnosis of invasive wound mucormycosis and initiation of treatment [53]. Molecular methods also have shown promise in detecting infections in tissues in immunocompromised patients much earlier, approximately 3–68 days before identification by conventional methods, via assays that targeted *Lichtheimia corymbifera*, *Mucor/Rhizopus*, and *Rhizomucor* 18S ribosomal RNA genes [54]. Polymerase chain reaction (PCR) assays could be practical screening tools in high-risk patients, enabling early therapeutic interventions.

Further, real-time polymerase chain reaction (qPCR) assays targeting Mucorales in blood/serum samples have easy sampling, low turnaround time (3 h), and can detect infections earlier (8 days) compared with conventional methods or radiological diagnosis [53,54,55]. A study by Baldin et al. has shown promising results with a sensitivity of 90% and specificity of 100% in detecting Mucorales-specific spore coat protein homolog CotH genes from urine samples in murine models [56]. However, across different studies, there is a wide range of variation in the target genes, extraction methodology, and type of PCR system used for diagnosis, which has limited assessment of the comparability of assays of this rare disease [57]. The recently introduced commercial real-time PCR assay can address most of these issues. However, a recent study found these assays had lower positivity than the in-house assay [58]. Further research and studies are needed to confirm the findings of this study and the utility of molecular methods for diagnosing mucormycosis.

**Next-generation sequencing**: Metagenomic next-generation sequencing (NGS), which can detect multiple organisms in a sample without the need for culture, has also been used to diagnose mucormycosis in blood and other body fluids. In a study involving 73 suspected mucormycosis cases, the assay demonstrated a positive predictive value of 72.6% [59]. This method’s major disadvantage is its lack of universal availability and the need for additional studies to clarify its role in mucormycosis diagnosis.

**Whole genome sequencing**: Whole genome sequencing of mucoralean fungi has the potential to aid in the identification of fungal outbreaks and has been used in the past in France to clarify the outbreak of mucormycosis due to different environmental strains [60].

## 6. Management of Mucormycosis

The effective management of mucormycosis poses significant challenges, necessitating a multimodal approach. A pivotal milestone is the control of mucormycosis risk factors, such as correcting metabolic abnormalities in diabetic patients and excising all infected tissues. This should be followed by the early administration of an active antifungal agent at an optimal dosage, with potential adjunctive therapies, as summarized in Table 1. Similarly, corticosteroids and immunosuppressive drugs must be used at the lowest possible dose. Early diagnosis is crucial for mucormycosis management, preventing progressive tissue invasion and its severe consequences, thereby enhancing outcomes and survival [61,62,63].

Previously, guidelines were predominantly available for specific populations or geographic regions and lacked comprehensive clinical, radiological, pathological, and histological details necessary for adequate mucormycosis treatment [61]. Addressing this gap, the European Confederation of Medical Mycology (ECMM) and the Mycoses Study Group Education and Research Consortium (MSG ERC) present a guidance document to facilitate clinical decision-making. This document overcomes previous limitations, providing new recommendations for diagnosis and treatment [7].

Liposomal amphotericin B (L-AmB) is strongly recommended as a first-line treatment at a full dose (5–10 mg/kg) from the onset of therapy [7,61,62]. It should commence as soon as possible, as delayed administration is associated with increased mortality [7,62,63]. Treatment must persist until a complete response is observed in radiological controls and a permanent regression of immunosuppressive status occurs [7,63]. Amphotericin B lipid complex (ABLC), another lipid formulation, is an option for mucormycosis without central nervous system involvement. At the same time, the use of AmB deoxycholate (AmBd) is strongly discouraged due to associated toxicity (Table 1).

Isavuconazole (ISZ), available in oral or IV formulations, is a recent drug approved for mucormycosis treatment [64,65,66]. Compared with older drugs, it presents advantages like linear pharmacokinetics, lower drug interactions, lower QT prolongation, no need for dose adjustment in kidney, liver failure, or obesity, and excellent oral bioavailability [64,66]. Despite lower in vitro activity than posaconazole, isavuconazole effectively decreases fungal burden and improves survival in a neutropenic mouse model of mucormycosis [64,66]. Notably, ISZ penetration of the blood–brain barrier in animal models is limited [64,66].

New formulations of posaconazole (PSZ), such as IV and delayed-release (DR) tablets, offer better bioavailability and increased drug exposure than previous oral solutions [61,67,68,69]. DR tablets provide less variability in absorption and are unaffected by food administration, making them moderately recommended compared with oral suspension, which are recommended marginally for mucormycosis [7,62,63]. However, the intravenous form is solubilized in cyclodextrin, which may lead to renal issues and need close monitoring [7,63]. Routine therapeutic drug monitoring (TDM) is advised, especially in cases of suspected toxicity, treatment failure, drug interactions, obesity, or after a switch from IV to PO therapy [7,38,61].

Several new antifungal drugs are pending clinical approval, including rezafungin, ibrexafungerp, olorofim, and ancochleated amphotericin B. Their efficacy against Mucorales varies. Other antifungal drugs active against Mucorales in murine models but untested in humans include VT-1161 (oteseconazole), SCH 42427 (saperaconazole), APX001A (fosmanogepix), and PC1244.

Limited data exist on antifungal combination therapy, such as polyenes with azoles or polyenes plus echinocandins. The potential benefits of combination therapy must be further demonstrated in clinical trials. New topical formulations, like aerosolized L-AmB, show promise for local intrapulmonary drug delivery, decreasing systemic absorption and improving survival when administered early. There are anecdotal reports on the use of AmB eye drops for keratitis, oral AmB for gastrointestinal mucormycosis, intradiaphyseal incorporation cement beads in osteomyelitis, intrathecal administration in cerebral abscess, and percutaneous injection in cutaneous lesions [7,70,71,72,73,74]. The optimal usage of these formulations in conjunction with systemic therapy is yet to be determined [7].

Similarly, there is a lack of pediatric data, and no specific dosing recommendations for treating children currently exist as far as management is concerned [75]. Amphotericin B deoxycholate is better tolerated in children than adults [75,76]. Posaconazole is yet to be approved by the FDA below 13 years of age, and no specific dose recommendations exist; most of its use in this age group is off-label [77]. Treatment options are limited in patients with breakthrough mucormycosis, with amphotericin B emerging as the preferred therapeutic approach in these challenging cases [78].

In conclusion, the current ECMM MSG ERC recommendations prioritize surgical debridement with clean margins and liposomal amphotericin B as the first-line drug for mucormycosis therapy [7]. New formulations of ISZ and PSZ are considered for second-line treatment following L-AmB. Evidence for adjunctive therapies is scarce, and doubts on their effectiveness persist due to a lack of randomized prospective controlled studies. Ongoing developments in mucormycosis treatment include drugs under development, whose utility in clinical practice needs to be proven by clinical trials prior to incorporation into guidelines [63].

## Figures and Tables

**Figure 1 diagnostics-14-00314-f001:**
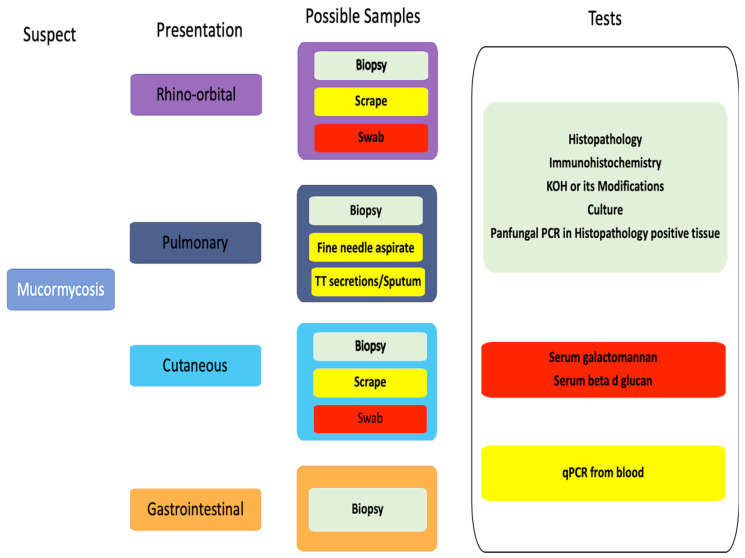
Diagnostic pathways in different clinical presentations of mucormycosis. Green—recommended, yellow—use with caution, red do not use.

**Table 1 diagnostics-14-00314-t001:** Recommended drug therapy for the treatment of mucormycosis.

	Antimicrobial Agents [7]		Comments
	Primary	Alternative	
Mucormycosis due to *Rhizopus*, *Rhizomucor*, *Lichtheimia*, or other Mucorales.	Liposomal amphotericin B (L-AmB) 5–10 mg/kg/day or Amphotericin B 1–1.5 mg/kg/day	Isavuconazole 200 mg IV/PO × 3/on day 1–2 and then 200 mg IV/PO/day or Posaconazole delayed-release (DR) tablets/IV 300 mg q12 × 2 doses as a loading dose, then 300 mg/day	Surgical excision of infected tissue, wherever feasible, is crucial for survival. Combination therapy: the addition of an echinocandin with liposomal amphotericin B appears to have a greater safety profile Treatment must be continued until: - resolution of signs and symptoms - resolution or improvement of the radiological picture - resolution of immunocompromise

## Data Availability

No new data were created or analyzed in this study. Data sharing is not applicable to this article.

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
