# Peer review of "From Spores to Solutions: A Comprehensive Narrative Review on Mucormycosis"

_diagnostics, 2024, doi:10.3390/diagnostics14030314_

Round 1

Reviewer 1 Report

Comments and Suggestions for Authors

I have read the review article "From Spores to Solutions: A Comprehensive Narrative Review on Mucormycosis." The review is interesting for the clinician and is well-written. There are some minor problems:

-Some acronyms are not adequately clarified in the text, such as ROCM (page 3)

-The last sentence of the first paragraph on page 5 ("low plasma concentrations of posaconazole...") is repetitive.

Despite this, I believe that the article can be published.

Author Response

We thank the reviewer for his encouraging comments. 

I have read the review article "From Spores to Solutions: A Comprehensive Narrative Review on Mucormycosis." The review is interesting for the clinician and is well-written. There are some minor problems: Point by point reply to queries raised can be found below:

Comment: Some acronyms are not adequately clarified in the text, such as ROCM (page 3)-

Reply:  All acronyms have been clarified in the text e.g. Rhino-orbito-cerebral mucormycosis (ROCM)  on page 3

Comment: The last sentence of the first paragraph on page 5 ("low plasma concentrations of posaconazole...") is repetitive.

Reply: The sentence has been revised to "Kang et al. and Lebeaux suggest varying sensitivity to posaconazole among Mucorales and low plasma concentrations of posaconazole due to diarrhea and mucositis in these patients further contribute to breakthrough mucormycosis [37,38]." 

Reviewer 2 Report

Comments and Suggestions for Authors

Nice review. There are a couple of issues that should be addressed.

1. There should be a section devoted to mucormycosis assocu=iated with COVID similar to those listed under clinical manifectations.

2. Under cutaneous mucormycosis, it should be clarified that this is primary infection rather than a manifestation of hematogenous dissemination. It might be helpful to title this section as Primary Cutaneous Mucormycosis.

3. When mentioning the Breath Test please specify if it may be used to diagnose non-pulmonary infections or is it limited to pulmonary infections only.

4. Under molecular methods, whole genome sequencing or next generation sequencing should also be mentioned.

5. Pg 8, under management, when mentioning other antifungal drugs , please indicate the assigned name if available: VT-1161 (oteseconazole); APX001A (manogepix).

Author Response

We thank the reviewer for his comments which have improved the quality and added value to the manuscript.

Comment: 1 There should be a section devoted to mucormycosis associated with COVID, similar to those listed under clinical manifestations.

Reply: A separate section on COVID-associated mucormycosis can now be found under clinical presentations " 

COVID-19 Associated Mucormycosis The highest number of cases of CAM were recorded in India[5,15,17]. In a study on 958 CAM cases, DKA and steroid overuse were the commonest underlying risk factors [5,28]. All the clinical presentations mentioned above have been observed in patients with CAM. ROCM was the most common presentation worldwide in patients with CAM and had lower mortality (14% or more) compared to other presentations like pulmonary(70-80%), primary cutaneous(50%), gastrointestinal(76%) and disseminated mucormycosis(76%) [15,17,28]." 

Comment 2: Under cutaneous mucormycosis, it should be clarified that this is a primary infection rather than a manifestation of hematogenous dissemination. It might be helpful to title this section as Primary Cutaneous Mucormycosis.

Reply: The title of this section has been changed to Primary Cutaneous Mucormycosis. The first sentence has been revised to “ Primary Cutaneous Mucormycosis occurs following direct percutaneous inoculation of fungal spores and is not due to dissemination from another site.” To clarify that this is a primary infection rather than a manifestation of hematogenous dissemination.

Comment 3: When mentioning the Breath Test, please specify if it may be used to diagnose non-pulmonary infections or if it is limited to pulmonary infections only.

Reply: The breath test is a prototype not yet commercially or clinically available; its source was an abstract in the ID week. Hence, the paragraph on the breath test has been removed. This helped us accommodate the new headings on CAM, mNGS, and WGS.

Comment 4: Under molecular methods, whole genome sequencing or next-generation sequencing should also be mentioned.

Reply: The following has been added to the manuscript “ Next-generation sequencing: Metagenomic next-generation sequencing (NGS), which can detect multiple organisms in a sample without the need for culture, has also been used to diagnose mucormycosis in blood and other body fluids. In a study involving 73 suspected mucormycosis cases, the assay demonstrated a positive predictive value of 72.6% [59]. This method's major disadvantage is its lack of universal availability and the need for additional studies to clarify its role in mucormycosis diagnosis.

Whole genome sequencing: Whole genome sequencing of mucoralean fungi has the potential to aid in the identification of fungal outbreaks and has been used in the past in France to clarify the outbreak of mucormycosis due to different environmental strains [60].”

Comment: 5. Pg 8, under management, when mentioning other antifungal drugs , please indicate the assigned name if available: VT-1161 (oteseconazole); APX001A (manogepix).

Reply:  The line has been revised to “ Other antifungal drugs active against Mucorales in murine models but untested in humans include VT-1161(oteseconazole), SCH 42427(saperaconazole), APX001A(fosmanogepix), and PC1244” to indicate the assigned name where available